# Federated Sparse Training: Lottery Aware Model Compression for Resource Constrained Edge

## Abstract

Limited computation and communication capabilities of clients pose significant challenges in federated learning (FL) over resource-limited edge nodes. A potential solution to this problem is to deploy off-the-shelf sparse learning algorithms that train a binary sparse mask on each client with the expectation of training a consistent sparse server mask. However, as we investigate in this paper, such naive deployments result in a significant accuracy drop compared to FL with dense models, especially under clients' low resource budgets. In particular, our investigations reveal a serious lack of consensus among the trained masks on clients, which prevents convergence on the server mask and potentially leads to a substantial drop in model performance. Based on such key observations, we propose *federated lottery aware sparsity hunting* (FLASH), a unified sparse learning framework to make the server win a lottery in terms of a sparse sub-model, which can greatly improve performance under highly resource-limited client settings. Moreover, to address the issue of device heterogeneity, we leverage our findings to propose *hetero-FLASH*, where clients can have different target sparsity budgets based on their device resource limits. Extensive experimental evaluations with multiple models on various datasets (both IID and non-IID) show superiority of our models in yielding up to $\sim 10.1\%$ improved accuracy with $\sim 10.26\times$ fewer communication costs, compared to existing alternatives, at similar hyperparameter settings.

## 1 Introduction

Federated learning (FL) [30] is a popular form of distributed training, which allows multiple clients to learn a shared global model without the requirement to transfer their private data. However, clients' heterogeneity and resource limitations pose significant challenges for FL deployment over edge nodes, including mobile phones and IoT devices. To resolve these issues, various methods have been proposed over the past few years including efficient learning for heterogeneous collaborative training [27, 42], distillation [12], federated dropout techniques [15, 4], efficient aggregation for faster convergence and reduced communication [34, 25]. However, these methods do not necessarily address the growing concerns of highly computation and communication limited edge.

Meanwhile, reducing the memory, compute, and latency costs for deep neural networks in centralized training is an active area of research. In particular, recently proposed *sparse learning* strategies [8, 20, 31, 5, 33] effectively train weights and associated binary *sparse masks* to allow only a fraction of model parameters to be updated during training, potentially enabling the lucrative reduction in both the training time and FLOPs [32, 33], while creating a *model to meet a target parameter density denoted as d, and is able to yield accuracy close to that of the unpruned baseline.*

However, the challenges and opportunities of sparse learning in FL is yet to be fully unveiled. Only very recently, few works [2, 16] have tried to leverage sparse learning in FL primarily to show their efficacy in non-IID settings. Nevertheless, these works primarily used sparsity for non-aggressive

Submitted to 36th Conference on Neural Information Processing Systems (NeurIPS 2022). Do not distribute.

model compression, limiting the actual benefits of sparse learning, and assumed multiple local epochs, that may further increase the training time for stragglers making the overall FL process inefficient [40]. Moreover, the server-side pruning used in these methods may not necessarily adhere to the layers' pruning sensitivity[1] [7] that often plays a crucial role in sparse model performance [20, 39, 35]. Another recent work, ZeroFL [32], has explored deploying sparse learning in FL settings with limited client epochs. However, [32] could not leverage any advantage of model sparsity in the clients' down-link communication cost and had to keep significantly more parameters active compared to a target $d$ to yield good accuracy. Moreover, as shown in Fig. 1(b), for $d = 0.05$, ZeroFL still suffers from substantial accuracy drop of $\sim 14\%$ compared to the baseline.

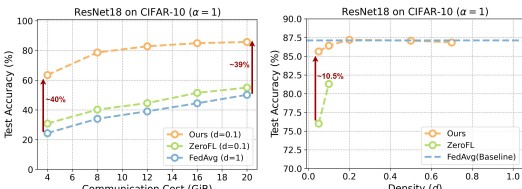

Figure 1: Comparison of (a) accuracy at different communication budget with, ZeroFL [32] and FedAvg. (w/ $d = 1.0$) (b) Accuracy vs. parameter density of each client. Proposed approach can significantly outperform the existing alternative [32] at ultra-low target parameter density ($d$).

**Our Contributions.** Our contribution is fourfold. In view of the above limitations, we first identify crucial differences between a centralized and the corresponding FL model, in learning the sparse masks for each layer. In particular, we observe that in FL, the server model fails to yield convergent sparse masks. In contrast, the centralized model show significantly higher convergence trend in learning sparse masks for all layers. We then experimentally demonstrate the utility of pruning sensitivity and mask convergence in yielding good accuracy setting the platform to close the performance gap in sparse FL.

We then leverage our findings and present *federated lottery aware sparsity hunting* (FLASH), a sparse FL methodology addressing the aforementioned limitations. At the core, FLASH leverages a two-stage FL, a robust and low-cost layer sensitivity evaluation stage which identifies a good predefined sparse mask for the clients and a training stage. We claim the first stage to play a key role in communication-efficient learning of a model which yields SOTA accuracy in sparse FL.

To deal with resource heterogeneity, we further extend our methodologies to *hetero*-FLASH, where we assume a critical scenario of individual clients having different $d$. Here, to deal with the unique problem of the server selecting different sparse models for clients, we present server-side gradual mask sub-sampling, that identifies sparse masks via a form of layer sensitivity re-calibration, starting for models with highest to that with lowest density support.

We conduct extensive experiments on MNIST, FEMNIST, and CIFAR-10 with different models for both IID and non-IID client data partitioning. Experimental results show that, compared to the existing alternative [32], at iso-hyperparameter settings, FLASH can yield up to $\sim 8.9\%$ and $\sim 10.1\%$, on IID and non-IID data distribution of CIFAR-10 dataset, respectively, with reduced communication of up to $\sim 10.2\times$ (Table 3).

## 2 Related Works

**Model Pruning.** Over the past few years, a plethora of research has been done to perform efficient model compression via pruning, particularly in centralized training [29, 9, 28, 38, 14]. Pruning essentially identifies and removes the unimportant parameters to yield compute-efficient inference models. More recently, sparse learning [8, 20, 5, 33], a popular form of model pruning, has gained significant traction due to its popularity in yielding FLOPs advantage and potential speed-up even during training. In particular, it ensures only $d\%$ of the model parameters remain non-zero during the training for a target parameter density $d$, potentially enabling training complexity reduction.

**Dynamic network rewiring (DNR).** We leverage DNR [20], to learn the sparsity mask of each client. In DNR, a model starts with randomly initiated mask following the target parameter density $d$. After an epoch, the client evenly prunes the lowest $p_r\%$ weights from each layer based on absolute magnitude, where $p_r$ is prune rate. Note, this $p_r\%$ pruning happens on top of the sparse model with density $d$, allowing $p_r\%$ weights to be regrown. DNR then ranks each layer based on the normalized contribution of the summed non-zero weight magnitudes. Finally, the client regrows total $p_r\%$ weights in a non-uniform way, allowing more regrow to the layers having higher rank. This

---

[1]A layer with higher sensitivity demands higher % of non-zero weights compared to a less sensitive layer.

process iteratively repeats over epochs to finally learn the mask. **Federated learning for resource and communication limited edge.** To address device heterogeneity, existing works have explored the idea of heterogeneous training [15, 6, 37] allowing different clients to train on different fractions of full-model based on their compute-budget. On a parallel track, various optimizations are proposed in FL training framework to yield faster convergence, thus requiring fewer communication rounds [11, 10, 41, 26, 34, 17, 1].

A few research have leveraged pruning in FL [23, 18, 24]. In particular, in LotteryFL [23] and PruneFL [18], clients need to send the full model to the server regularly costing bandwidth. Moreover, in [23], each client trains a personalized mask to maximize the performance only on the local data.

Only a few contemporary works [16, 2, 32] tried to leverage the benefits of sparse learning in federated settings. In particular, [16] relied on a randomly initialized sparse mask, and recommended keeping it frozen [21] throughout the training, yet failed to provide any supporting intuition. FedDST [2], on the other hand, leveraged the idea of RigL [8] to perform sparse learning of the clients and relied on magnitude pruning at the server-side that does not necessarily adhere to the layer sensitivity towards a target density. Moreover, both the approaches assumed all clients can support a fixed $d$, a large number of local epochs, and focused primarily on only highly non-IID data without targeting ultra-low density $d$. More importantly, neither of these works investigated the key differences between centralized and FL sparse learning. With similar philosophy as ours, ZeroFL [32] first identified a key aspect of sparse learning in FL in terms of all clients' masks to be within $30\%$ of the total model weights to yield good accuracy at high compression. However, ZeroFL suffered significantly in failing to exploit a proportional advantage in communication saving as even for low parameter density $d$, all clients had to download the dense model and send back at least a model with $d = 0.3$. Furthermore, these algorithms sacrifice significant accuracy at ultra-low $d$.

# 3 Revisiting Sparse Learning: Why Does it Miss the Mark in FL?

Sparse learning uses proxies, including normalized momentum and normalized values of the non-zero weights [20, 5], to decide the layers and weights that are more sensitive towards pruning and update the binary sparse mask accordingly. Note, centralized training has shown significant benefits with sparse learning with FLOPs reduction during forward operations [8], and potential training speed-up of up to $3.3\times$ [32] while maintaining close to the baseline accuracy, even at $d \leq 0.1$. We now use a sparse learning, namely [20], in FL settings (refer to Table 1 for details) on CIFAR-10, where each client separately performs [20] to train a sparse ResNet18 and meet a fixed parameter density $d$, starting from a random sparse mask. After sending the updates to server, it aggregates them using FedAvg. We term this as *naive sparse training* (NST).

Table 1: FL training settings considered in this work.

| Dataset | Model | #Params. | Data-partioning | Rounds $(T)$ | Clients $(C_N)$ | Clients/Round $(c_r, c_d)$ | Optimizer | Aggregation type | #Local epochs$(E)$ | Sensitivity warmup$(E_d)$ | Batch Size |
|---|---|---|---|---|---|---|---|---|---|---|---|
| MNIST | MNISTNet | 262K | LDA | 400 | 100 | 10, 10 | | | | | 32 |
| CIFAR-10 | ResNet18 | 11.2M | | 600 | | | SGD | FedAvg [30] | 1 | 10 | |
| FEMNIST | Same as [3] | 6.6M | [34] | 1000 | 3400 | 34, 34 | | | | | 16 |

**Observation 1.** *At high compression $d \leq 0.1$, the collaboratively learned FL model significantly sacrifices performance, while the centralized sparse learning yields close to baseline performance.*

As shown in Fig.2(a), naive deployment of sparse learning significantly sacrifices accuracy in FL. In particular, for $d = 0.1$, the trained server-side model suffers an accuracy drop of $3.67\%$. At even lower $d = 0.05$, this drop significantly increases to $12.03\%$, hinting at serious limitations of sparse learning in FL. However, in centralized sparse learning, the model yields close to the baseline accuracy, even at $d = 0.05$.

**Observation 2.** *As the training progresses, the sparse masks in centralized training tend to agree across epochs, showing convergence, while server mask in FL does lack agreement across rounds.*

**Definition 1. Sparse mask mismatch.** For a model at round $t$, we define the *sparse mask mismatch* (SM) $\mathtt{sm}^t$ as the Jaccard distance that is measured as follows.

$$\mathtt{sm}^t = 1 - \frac{(\sum_{l=1}^{L} \mathcal{M}_l^t \cap \mathcal{M}_l^{t-1})}{(\sum_{l=1}^{L} \mathcal{M}_l^t \cup \mathcal{M}_l^{t-1})} \tag{1}$$

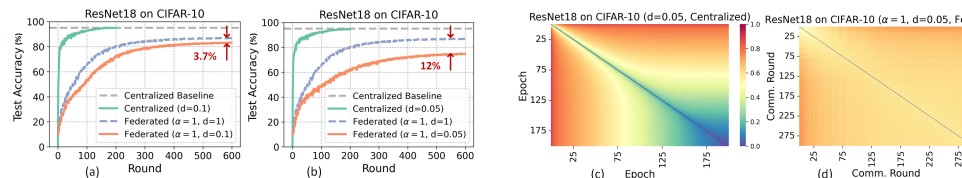

Figure 2: (a-b) Accuracy vs. round plot on deployment of off-the-shelf [20] sparse learning in FL for different $d$, (c-d) visualization of the Model's SM in terms of Jaccard distance while training with sparse learning for (c) centralized and (d) FL, respectively.

where $\mathcal{M}_l^t$ represents the sparse mask tensor for layer $l$ at the end of round $t$. Interestingly, as depicted in Fig. 2(a), the SM for centralized learning tends to zero as the training progresses. In contrast, with the same model, dataset and $d$ values, in FL, the SM remains $> 0.4$ indicating a substantial distinction in the sparse mask learning between centralized and federated learning.

# 4  FLASH: Methodology

To win a lottery of having a sparse network yielding high accuracy at reduced parameters, we identify a key characteristic of sparse learning, the pruning sensitivity. To explicitly adhere to this important aspect, in FLASH, we present a two-stage sparse FL method, **stage1:** targeting sensitivity analysis to identify good initial sparse mask for each layer, **stage2:** targeting training to weights. In particular, to evaluate layer sensitivity in stage1, the server randomly selects a small fraction of clients ($[\mathcal{C}_d]$), each locally sparse learning [20] for few warm-up epochs ($E_d$) ($L4$-$8$ in Algo. 1). Upon collection of layer-wise sensitivity from the clients, for each layer $l$, the server calculates the average sensitivity per layer[2] $\hat{d}^l$ as $\frac{\sum_{i=1}^{c_d} d_i^l}{c_d}$, where $d_i^l$ is the density at layer $l$ in $i^{th}$ client. As these averaged layer-wise density values may not necessarily yield to the target density $d$, for a model with $K$ parameters we follow the following *density re-calibration*

$$d_c^l = \hat{d}^l . r_f, \text{ where } r_f = \frac{d \times K}{\sum_{l=1}^{L} \hat{d}^l . k^l} \tag{2}$$

$k^l$ is dense model's parameter size for layer $l$. For each layer $l$ of the model, the server then creates a binary sparse mask tensor that is randomly initialized, with a fraction of 1s $\propto d_c^l$ ($L9$). In stage2, at each round, the clients train the model for $E$ epochs ($L19$-$24$) with a mask frozen ($L22$).

However, in FL settings, the masks often show poor convergence (§3, Obs. 2). To address this, in stage 2, we present the idea of sparse FL with pre-defined layer masks at initialization ($L13$). The frozen mask allows all the clients to have a forcefully convergent mask ($\mathsf{sm}^t = 0$ for all $t$). Moreover, as FLASH disentangles the sensitivity evaluation stage from the training, the pre-defined mask in this scenario benefits from the notion of layer sensitivity. Interestingly, earlier research [2] hinted at poor model performance with pre-defined masks, contrasting ours where we see significantly improved model performance, implying the importance of stage1 (as will be elaborated in §5).

**Extension to support heterogeneous parameter density.** To support different density budgets for different clients, we now present hetero-FLASH. Let us assume a total of $N$ support densities $d_{set} = [d_1, .., d_M]$, where $d_i < d_{i+1}$. First, we perform a sensitivity warm-up, to create the masks for the clients' with the highest density $d_N$. For any other density $d_i$, a sparse mask is subsampled from the mask with density $d_{i+1}$. Note, while creating the mask from $d_{i+1}$ to $d_i$, we follow the layer-wise density re-calibration approach as mentioned earlier. In hetero-FLASH, server aggregates the update following *weighted fedAvg* (WFA) instead of fedAvg. In particular, with similar inspiration as [6], to give equal importance to each parameter update in such heterogeneous settings, WFA averages the values by their number of non-zero occurrences among the participating clients.

# 5  Experiments

**Datasets and Models.** We evaluated the performance of FLASH on MNIST[22], Federated EMNIST (FEMNIST) [3], and CIFAR-10 [19] datasets with the CNN models described in [30], [3], and ResNet18, respectively. For data partitioning of MNIST and CIFAR-10, we use Latent Dirichlet

---

[2]For a sparse model it is evaluated as the ratio $\frac{\texttt{\# of non-zero layer parameters}}{\texttt{\# layer parameters}}$ [7].

**Algorithm 1:** FLASH Training.

**Data:** Training rounds $T$, local epochs $E$, client set $[\mathcal{C}_N]$, clients per rounds $c_r$, target density $d$, sensitivity warm-up epochs $E_d$, density warm up client count $c_d$, initial value of freeze masks $m_{freez} = 0$ and aggregation type $t_{aggr}$.

```
1  M^init ← createRandomMask(d)
2  Θ^init ← initMaskedWeight(M^init)
3  serverExecute:
4  Randomly sample c_d clients [C_d] ⊂ [C_N]
5  for each client c ∈ [C_d] in parallel do
6  │    Θ_c ← clientExecute(Θ^init, E_d, 0) # m_freeze = 0
7  │    S_c ← computeSensitivity(Θ_c)
8  end
9  M^0 ← initMask([S_c], d)
10 Θ^0 ← initMaskedWeight(M^0)
11 m_freez ← 1
12 for each round t ← 1 to T do
13 │    Randomly sample c_r clients [C_r] ⊂ [C_N]
14 │    for each client c ∈ [C_r] in parallel do
15 │    │    Θ_c^t ← clientExecute(Θ^{t-1}, E, m_freez)
16 │    end
17 │    Θ^t ← aggrParam([Θ_c^t], t_aggr)
18 end
19 clientExecute(Θ_c, E, m_freez):
20 Θ_{c0} ← Θ_c
21 for local epoch i ← 1 to E do
22 │    Θ_{ci} ← doSparseLearning(Θ_{ci-1}, m_freez)
23 end
24 return Θ_{cE}
```

Table 2: Results with FLASH and its comparison with NST and PDST.

| Dataset | Data Distribution | Density (d) | Baseline Acc % | NST Acc % | PDST Acc % | FLASH Acc % |
|---|---|---|---|---|---|---|
| MNIST | IID ($\alpha = 1000$) | 1.0 | $98.79 \pm 0.06$ | – | – | – |
| | | 0.1 | – | $97.57 \pm 0.11$ | $97.09 \pm 0.18$ | $\mathbf{98.21 \pm 0.06}$ |
| | | 0.05 | – | $95.19 \pm 0.56$ | $94.8 \pm 1.04$ | $\mathbf{97.46 \pm 0.14}$ |
| | non-IID ($\alpha = 1.0$) | 1.0 | $98.76 \pm 0.06$ | – | – | – |
| | | 0.1 | – | $97.36 \pm 0.19$ | $96.82 \pm 0.25$ | $\mathbf{97.96 \pm 0.13}$ |
| | | 0.05 | – | $95.75 \pm 0.31$ | $95.34 \pm 0.77$ | $\mathbf{97.3 \pm 0.26}$ |
| | non-IID ($\alpha = 0.1$) | 1.0 | $98.45 \pm 0.17$ | – | – | – |
| | | 0.1 | – | $96.19 \pm 0.22$ | $94.41 \pm 1.23$ | $\mathbf{97.22 \pm 0.43}$ |
| | | 0.05 | – | $91.66 \pm 1.74$ | $91.06 \pm 1.1$ | $\mathbf{95.7 \pm 0.37}$ |
| CIFAR-10 | IID ($\alpha = 1000$) | 1.0 | $88.56 \pm 0.06$ | – | – | – |
| | | 0.1 | – | $84.89 \pm 0.26$ | $86.72 \pm 0.09$ | $\mathbf{88 \pm 0.28}$ |
| | | 0.05 | – | $77.48 \pm 0.54$ | $84.38 \pm 0.12$ | $\mathbf{86.99 \pm 0.14}$ |
| | non-IID ($\alpha = 1.0$) | 1.0 | $87.13 \pm 0.18$ | – | – | – |
| | | 0.1 | – | $83.46 \pm 0.19$ | $85.07 \pm 0.24$ | $\mathbf{86.42 \pm 0.49}$ |
| | | 0.05 | – | $75.1 \pm 0.76$ | $83.33 \pm 0.14$ | $\mathbf{85.64 \pm 0.58}$ |
| | non-IID ($\alpha = 0.1$) | 1.0 | $77.64 \pm 0.49$ | – | – | – |
| | | 0.1 | – | $71.18 \pm 1.23$ | $74.82 \pm 0.72$ | $\mathbf{76.74 \pm 1.46}$ |
| | | 0.05 | – | $61.29 \pm 2.76$ | $72.32 \pm 1.05$ | $\mathbf{75.47 \pm 2.31}$ |
| FEMNIST | non-IID | 1.0 | $84.68 \pm 0.20$ | – | – | – |
| | | 0.1 | – | $76.92 \pm 0.42$ | $76.01 \pm 1.26$ | $\mathbf{82.70 \pm 0.26}$ |
| | | 0.05 | – | $61.9 \pm 2.6$ | $63.65 \pm 0.86$ | $\mathbf{81.18 \pm 0.36}$ |

Allocation (LDA)[34] with three different $\alpha$ ($\alpha = 1000$ for IID and $\alpha = 1$ and $0.1$ for non-IID). For FEMNIST, we employ the same setting as in [11], which partitions the data based on the writer into 3400 clients, making it inherently non-IID.

**Training Hyperparameters.** We use Clients' starting learning rate ($\eta_{init}$) as 0.1 that is exponentially decayed to 0.001 ($\eta_{end}$) at the end of training. Specifically, learning rate for participants at round t is $\eta_t = \eta_{init}\left(\exp\left(\frac{t}{T}\log\left(\frac{\eta_{init}}{\eta_{end}}\right)\right)\right)$. In all the sparse learning experiments, prune rate is set to $0.25^3$. Summary of the rest of the hyperparameters can be found in 1. Furthermore, we report the final results as the averaged accuracy with corresponding std of three different seeds in the tables.

## 5.1 Experimental Results with FLASH

To understand the importance of stage1 in FLASH methodology, we identify a baseline training with uniform layer sensitivity driven *pre-defined sparse training* (PDST) in FL. Table 2 details the performance of FLASH at different levels of $d$, for various choices of sparse learning methods. In particular, as we can see in Table 2 column 5 and 6, the performance of both NST and PDST produced models cost heavy accuracy drop at ultra low parameter density $d = 0.05$. For example, on CIFAR-10 ($\alpha = 0.1$), models from NST and PDST sacrifice an accuracy of $16.35\%$ and $5.32\%$, respectively.

---

[3]Prune rate controls the fraction of non-zero weights participating in the redistribution during sparse learning.

Table 3: Comparison of FLASH on various performance metrics with existing alternative sparse federated learning schemes.

| Dataset | Data Distribution | Method | Density | Acc% | Down-link Savings | Up-link Savings |
|---|---|---|---|---|---|---|
| CIFAR-10 | IID | ZeroFL [32] | 0.1 | $82.71 \pm 0.37$ | $1\times$ | $1.6\times$ |
| | | FLASH (ours) | 0.1 | $\mathbf{88 \pm 0.28}$ | $\mathbf{9.8\times}$ | $\mathbf{9.8\times}$ |
| | | ZeroFL [32] | 0.05 | $78.22 \pm 0.35$ | $1\times$ | $1.9\times$ |
| | | FLASH (ours) | 0.05 | $\mathbf{86.99 \pm 0.14}$ | $\mathbf{19.5\times}$ | $\mathbf{19.5\times}$ |
| | non-IID ($\alpha = 1.0$) | ZeroFL [32] | 0.1 | $81.04 \pm 0.28$ | $1\times$ | $1.6\times$ |
| | | FLASH (ours) | 0.1 | $\mathbf{86.42 \pm 0.49}$ | $\mathbf{9.8\times}$ | $\mathbf{9.8\times}$ |
| | | ZeroFL [32] | 0.05 | $75.54 \pm 1.15$ | $1\times$ | $1.9\times$ |
| | | FLASH (ours) | 0.05 | $\mathbf{85.64 \pm 0.58}$ | $\mathbf{19.5\times}$ | $\mathbf{19.5\times}$ |
| FEMNIST | non-IID | ZeroFL [32] | 0.05 | $77.16 \pm 2.07$ | $1\times$ | $\mathbf{17.7\times}$ |
| | | FLASH (ours) | 0.05 | $\mathbf{81.18 \pm 0.36}$ | $\mathbf{14.6\times}$ | $14.6\times$ |

Table 4: Performance of hetero-FLASH where support density set is $d_{set} = [0.1, 0.15, 0.2]$.

| Dataset | Data Distribution | Max Client Density | Hetero-FLASH Acc % |
|---|---|---|---|
| MNIST | IID ($\alpha = 1000$) | | $98.29 \pm 0.05$ |
| | non-IID ($\alpha = 1.0$) | 0.2 | $98.29 \pm 0.09$ |
| | non-IID ($\alpha = 0.1$) | | $97.63 \pm 0.22$ |
| CIFAR-10 | IID ($\alpha = 1000$) | | $87.19 \pm 0.26$ |
| | non-IID ($\alpha = 1.0$) | 0.2 | $86.16 \pm 0.04$ |
| | non-IID ($\alpha = 0.1$) | | $75.23 \pm 1.26$ |
| FEMNIST | non-IID | 0.2 | $82.58 \pm 0.24$ |

However, at comparatively higher density ($d = 0.1$), both can yield models with a lower accuracy difference from the baseline by around $6.46\%$ and $2.82\%$. FLASH, on the other hand, can maintain **close to the baseline accuracy** at even ultra-low density for all data partitions. *These results clearly highlight the efficacy of both sensitivity driven spare learning (as FLASH > PDST) and early mask convergence in FL settings*. Moreover, as in FLASH, the clients' do not need to send the mask at all, *allowing us to yield proportional communication advantage as the model sparsity.*

**Comparison with ZeroFL.** Despite leveraging a form of sparse learning [33], ZeroFL required significantly higher up-link/down-link communication cost compared to the target density $d$. This enables FLASH to gain a significant advantage in communication saving over ZeroFL, particularly for FLASH, as it only asks for the reduced size parameters to be communicated between the server and clients. In particular, we evaluate the communication saving as the ratio of the dense model size and corresponding sparse model size with the tensors represented in compressed sparse row (CSR) format [36]. As depicted in Table 3[4], FLASH can yield an accuracy improvement of up to $10.1\%$ at a reduced communication cost of up to $10.26\times$ (computed at up-link when both send sparse models).

## 5.2 Experimental Results with Hetero-FLASH

Table 4 shows the performance of hetero-FLASH where the clients can have three possible density budgets as defined by the $d_{set}$. To train on all the density values, we split clients into three groups, each having $40\%$, $30\%$, and $30\%$ of total clients, and corresponds to density 0.2, 0.15, and 0.1, respectively. Then, every round, $10\%$ from each group is sampled to participate in training the model.

## 6  Conclusions

This paper presented federated lottery-aware sparsity hunting methodologies to yield low parameter density server models with insignificant accuracy drop compared to the dense counterparts. In particular, we demonstrated an efficient sparse learning solution tailored for FL, enabling better computation and communication benefits over existing sparse learning alternatives. We experimentally showed the superiority of our model in yielding up to $\sim10.1\%$ improved accuracy with $\sim10.26\times$ fewer communication costs compared to the existing alternatives [32], at similar hyperparameter settings.

**Societal impact.** FLASH can efficiently learn low parameter FL models potentially reducing the energy budget thus carbon footprint of edge devices participating in FL.

---

[4]We understand for FEMNIST, ZeroFL reported significantly higher up-link saving, however, to the best of our understanding it should be similar to their report on other datasets, i.e. $\sim1.9\times$.

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

# 7 Supplementary

## 7.1 Model Architectures

Table 5 shows the model architectures used for MNIST and FEMNIST datasets. For CIFAR-10 we used ResNet18 [13] with the first `CONV` layer kernel size as $3 \times 3$ instead of original $7 \times 7$.

Table 5: Architecture used for MNIST and FEMNIST datasets

| MNIST | FEMNIST |
|---|---|
| `CONV`$5 \times 5(C_o = 10)$ | `CONV`$5 \times 5(C_o = 32)$ |
| `max_pool` | `max_pool` |
| `CONV`$5 \times 5(C_o = 20)$ | `CONV`$5 \times 5(C_o = 64)$ |
| `max_pool` | `max_pool` |
| `FC`$(5120, 50)$ | `FC`$(3136, 2048)$ |
| `FC`$(50, 10)$ | `FC`$(2028, 62)$ |

## 7.2 Hetero-FLASH Algorithm

Algorithm 2 details the training algorithm in hetero-FLASH.

---

**Algorithm 2:** Hetero-FLASH Training.

---

**Data:** Training rounds $T$, local epochs $E$, client set $[[\mathcal{C}_{N_1}], ..., [\mathcal{C}_{N_M}]]$, clients per rounds $c_r$, target density set $d_{set} = [d_1, ..., d_M]$, sensitivity warm-up epochs $E_d$, density warm up client count $c_d$, initial value of freeze masks $m_{freez} = 0$ and aggregation type $t_{aggr}$.

**1** $\mathcal{M}^{init} \leftarrow \texttt{createRandomMask}()$

**2** $\Theta^{init} \leftarrow \texttt{initMaskedWeight}(\mathcal{M}^{init})$

**3** $\underline{\texttt{serverExecute}}$:

**4** Randomly sample $c_d$ clients $[\mathcal{C}_d] \subset [\mathcal{C}_{N_M}]$

**5** **for** *each client* $c \in [\mathcal{C}_d]$ ***in parallel*** **do**

**6** $\quad$ $\Theta_c \leftarrow \texttt{clientExecute}(\Theta^{init}, E_d, 0)$

**7** $\quad$ $\mathcal{S}_c \leftarrow \texttt{computeSensitivity}(\Theta_c)$

**8** **end**

**9** $\mathcal{M}^0 \leftarrow \texttt{initMask}([\mathcal{S}_c], d_{set})$

**10** $\Theta^0 \leftarrow \texttt{initMaskedWeight}(\mathcal{M}^0)$

**11** $m_{freez} \leftarrow 1$

**12** **for** *each round* $t \leftarrow 1$ **to** $T$ **do**

**13** $\quad$ Randomly sample $c_r$ clients $[\mathcal{C}_r] \subset [\mathcal{C}_N]$

**14** $\quad$ **for** *each client* $c \in [\mathcal{C}_r]$ ***in parallel*** **do**

**15** $\quad\quad$ $\Theta_c^{t-1} \leftarrow applyClientMask(\Theta^{t-1}, c)$

**16** $\quad\quad$ $\Theta_c^t \leftarrow \texttt{clientExecute}(\Theta_c^{t-1}, E, m_{freez})$

**17** $\quad$ **end**

**18** $\quad$ $\Theta^t \leftarrow \texttt{aggrParam}([\Theta_c^t], t_{aggr})$

**19** **end**

**20** $\underline{\texttt{clientExecute}}(\Theta_c, E, m_{freez})$ :

**21** $\Theta_{c^0} \leftarrow \Theta_c$

**22** **for** *local epoch* $i \leftarrow 1$ **to** $E$ **do**

**23** $\quad$ $\Theta_{c^i} \leftarrow \texttt{doSparseLearning}(\Theta_{c^{i-1}}, m_{c^{freez}})$

**24** **end**

**25** return $\Theta_{c^E}$

---

