# OpenReview forum: "Federated Sparse Training: Lottery Aware Model Compression for Resource Constrained Edge"
_NeurIPS.cc/2022/Workshop/Federated_Learning — FL-NeurIPS 2022 Poster_

### Official Review · Reviewer_6LHy · 2022-10-15
**Review for Federated Sparse Training**

The authors propose a new communication-efficient federated learning (FL) framework in two stages. In the first stage, a selected group of clients perform sparse learning following [20] and communicates the learned layer-wise (pruning) sensitivity of each parameter to the server. Then, in the second stage, the server averages the layer-wise sensitivities from each client and constructs a binary mask using the average sensitivities. This mask is communicated to all the clients and kept frozen for the rest of the training. The clients then train the parameters using the frozen mask. This approach provides communication efficiency since the clients essentially train sparse networks. Experimental results on CIFAR-10, MNIST and FEMNIST show improvements over the baselines.

I think the paper has interesting observations and may lead to promising directions in future work. However, I believe there are some unclear statements and points that need to be addressed (perhaps in a longer version of the paper). I list my concerns below:

- The proposed strategy consists of two stages, where the first stage is for learning the mask and the second stage is for learning the parameters. I couldn't find details on how long the first stage takes and how many clients are needed to learn the mask. In the text, the authors say "a small fraction of clients" and " for few warm-up epochs", but I believe we need a more precise description of these quantities to better understand whether stage 1 is extremely costly or not. In one extreme, if the number of warm-up epochs is too large, then the improvements of the proposed framework could be mostly from the sparse learning strategy [20] used in stage 1. In that case, the proposed framework would be more like an application of the lottery ticket hypothesis in FL, where the tickets are found via [20] at the first stage, which is computationally very expensive. So, I think to make their claims stronger, the authors should give more details on stage 1.

- Also, it is not clear if the number of epochs at stage 1 is considered in the comparisons against the baselines. For a fair comparison, I believe the authors should make sure that $$\text{number of epochs in baselines} = \text{number of warm-up epochs in stage1} + \text{number of epochs in stage 2}$$
It is not clear from the text whether or not the authors incorporated this.

- In Section 5.1, it says FLASH does not need to send the mask at all. I think it would be nice to make this claim more clear. I think the authors meant that the masks are never communicated in stage 2.  But they have to be communicated at least once at the end of stage 1.

- Again, in Section 5.1, the authors say the clients communicate the sparse model with tensors represented in compressed sparse row (CSR) format. Why is this necessary if the server and all the clients have the same frozen mask at all times in stage 2? Wouldn't it be much more efficient if the clients communicate a list of non-pruned parameters (or the gradients of them) since the server already knows their locations? Please correct me if I am missing something here. Is there an additional pruning in stage 2? If so, I don't think it was clear from the text.

Minor:

- I think it would be easier for the readers if baseline methods were re-explained in the table captions with references.

---

### Official Review · Reviewer_2ei5 · 2022-10-17
**FLASH presents a simple yet very effective mechanism to train highly sparse federated models offering large communication savings**

The Authors present FLASH, a framework that first constructs a sparsity mask from a subset of clients, re-calibrates it meet a pre-specified density ratio, freezes it and then it is used to mask parameters during in a given model. This masking translates into, primarily, communication costs savings (both downlink and uplink) but can also offer compute savings. This last aspect however wasn't investigated in the paper.

The paper is fairly well written and provide a clear analysis on how directly doing sparse FL doesn't yield as good results as you'd obtain in the centralised setting (Section 3).  The results section is presented well. Good job!

The Authors often compare against ZeroFL, a previously published work that although it leverages sparsity at high ratios (90%, 95%) using an alternative masking mechanism, its main objective is not (unlike FLASH's) saving communication costs but to reduce training costs. This is why ZeroFL guarantees that weights and activation tensors during training will have a pre-defined density (e.g. 0.1, 0.05) and as a result, both forward and backward propagation can be done with sparse-dense convolutions/multiplications (as oppose of the common dense-dense counterpart). In FLASH, the Authors correctly highlight the limitations of ZeroFL when it comes to communication savings -- which are small in general since the goal of ZeroFL is not to learn a sparse model but to train a standard/dense model with highly sparse operations. For the sake of fairness and completeness, I would encourage the Authors to not only compare against ZeroFL in terms of communication costs, but also in terms of their acceleration potential (or at least mention this difference -- maybe at the end of line 114?): does FLASH uses sparse activations during backpropagation? If not, then the gradients w.r.t to the weights will be computed as a dense-dense op, limiting the acceleration capabilities.


Other minor comments:
*    Line 92, should there be a new paragraph for the **Federated learning for resource and communication limited edge** point?
*    Table 1 looks like it could read better with more spacing between columns (and there is some available)
*    Line 146 was a bit hard to read. I believe a "do" and an "a" are needed to be inserted

---

### Official Review · Reviewer_xp5t · 2022-10-18
**Review of the paper**

This paper presents federated lottery aware sparsity hunting (FLASH), an approach that identifies good mask in the training of clients through federated learning. The authors also extend the ideas to the heterogenous settings.

FLASH should algorithmic novelty by a two-stage assignment of the masks across clients. However, there are some concerns regarding the system implementation level that should be better justified.

Questions:
(1) FLASH aims at on-device settings where the hardware budget may be limited. However, it seems that adding the mask to the model does not save much in on-device model memory. Is it possible to justify more with profiling?

(2) How does FLASH performs forward and backward propagation? Does it compute the full gradients before adding the mask? How does the FLOPs saving comes from?

(3) Is it possible to specify the exact heterogeneous setting we have for hetero-FLASH. Does the client have to maintain different number of parameters in memory?

---

### Decision · Program_Chairs · 2022-10-20

Accept (Poster)